# Towards a Standardized Approach for the Geographical Traceability of Plant Foods Using Inductively Coupled Plasma Mass Spectrometry (ICP-MS) and Principal Component Analysis (PCA)

**DOI:** 10.3390/foods12091848

**Published:** 2023-04-29

**Authors:** Quang Trung Nguyen, Thanh Thao Nguyen, Van Nhan Le, Ngoc Tung Nguyen, Ngoc Minh Truong, Minh Tao Hoang, Thi Phuong Thao Pham, Quang Minh Bui

**Affiliations:** 1Center for Research and Technology Transfer, Vietnam Academy of Science and Technology, Hanoi 11353, Vietnam; nqtrung79@gmail.com (Q.T.N.); levannhan.na@gmail.com (V.N.L.); tungnguyen.vast@gmail.com (N.T.N.); anphuminh1011@gmail.com (N.M.T.); htm1205@gmail.com (M.T.H.); thaopham284@gmail.com (T.P.T.P.); 2Institute of Environmental Science and Public Health, Vietnam Union of Science and Technology Association, Hanoi 11353, Vietnam; thao7980@gmail.com; 3Faculty of Chemistry, Graduate University of Science and Technology, Vietnam Academy of Science and Technology, Hanoi 11353, Vietnam

**Keywords:** ICP-MS, multivariate statistic, principal component analysis, food geographical origin

## Abstract

This paper presents a systematic literature review focused on the use of inductively coupled plasma mass spectrometry (ICP-MS) combined with PCA, a multivariate technique, for determining the geographical origin of plant foods. Recent studies selected and applied the ICP-MS analytical method and PCA in plant food geographical traceability. The collected results from many previous studies indicate that ICP-MS with PCA is a useful tool and is widely used for authenticating and certifying the geographic origin of plant food. The review encourages scientists and managers to discuss the possibility of introducing an international standard for plant food traceability using ICP-MS combined with PCA. The use of a standard method will reduce the time and cost of analysis and improve the efficiency of trade and circulation of goods. Furthermore, the main steps needed to establish the standard for this traceability method are reported, including the development of guidelines and quality control measures, which play a pivotal role in providing authentic product information through each stage of production, processing, and distribution for consumers and authority agencies. This might be the basis for establishing the standards for examination and controlling the quality of foods in the markets, ensuring safety for consumers.

## 1. Introduction

Inductively coupled plasma mass spectrometry (ICP-MS) was first introduced by Houk et al. [1] and emerged commercially in the 1980s [2,3]. With its unique combination of desirable features, such as multielement capability, high sensitivity, low detection limits, wide linear dynamic range, high sample throughput, and isotopic discrimination, ICP-MS has rapidly become a valuable instrumental method for trace element analysis [4,5], particularly in the food industry [6]. Numerous studies have explored the use of ICP-MS for analyzing a variety of food matrices, including pork and swine meat [7,8], beef, bovine and veal [9,10], chicken [11,12], goat [11,13], mutton, lamb, and sheep [14,15], eggs [16,17], milk [18,19], fish [20,21], bivalve [22,23], oyster [24,25,26], as well as processed foods such as sugar, chocolate, Turkish delight, biscuits, butter, cheese, bread, pasta, vinegar, canned food, yogurt, juice, cucumar, and rice [27,28,29,30,31,32,33,34,35]. For a detailed summary of ICP-MS studies on the multielement analysis of plant-based food matrices using the PCA statistical method, refer to Table 1 below.

Despite the use of analytical ICP-MS methods to evaluate the content of elements in plant-based foods, relatively few publications exist on the metal content analyzed by ICP-MS in food. Most data analyzed by ICP-MS has been used for research on health risk assessment, daily uptake, and the development of analytical methods. Recently, multielement analytical data combined with principal component analysis has been increasingly applied in research on food geographical traceability [36,37]. The rapid development of ICP-MS applications in food traceability has made it a useful tool for verifying the origin of agricultural products and foods [38]. However, to date, no national or international standard methods for planting food geographical origin traceability have been established. Food traceability is still primarily conducted through the protected geographical indication (PGI) system, which involves identifying the specific ingredients of the food. However, analytical methods are mainly used to assess food quality rather than determine its origin. PGI is more of a commitment to agreement than a technical approach, and thus its application is limited in many countries.

**Table 1 foods-12-01848-t001:** Detected multielement in the plant-based food matrix.

Plant-Based Food Matrix	Elements	Range (mg.kg^−1^)	Ref
*F:* Peppers, tomato *N:* Wheat, corn, rice *T:* Potatoes, carrots	Micro: As, Pb, Cu, Cd, Zn, Sn, Hg	*F:* 0.002–1.896 *N:* <0.001–9.327 *T:* <0.001–2.37	[27]
*F:* golden berry *(Physalis peruviana)*, palm *(Euterpe edulis)*, acai *(Euterpe precatoria)* *N:* Brazil nut *(Bertholletia excels)*	Macro: ^44^Ca, ^39^K, ^24^Mg, ^31^P Micro: ^59^Co, ^65^Cu, ^60^Ni, ^85^Rb	Macro: *F:* 57.9–22,382 *N:* 544–26,180 Micro: *F:* 0.024–24 *N:* 1.1–191	[39]
*N:* Brown and white rice in South Korea	Macro: Al, Ba, Ca, Fe, K, Mg, P, S, Micro*:* As, Cd, Co, Cr, Cu, Mn, Ni, Pb, Sb, Se, Zn	Macro: 0.367–2103 Micro: 4 × 10^−4^-26	[40]
*F:* Black Pepper	Macro*:* K, Ca, Mg, Fe, Ba Micro: Mn, Cr, Ni, Ti, Cu, Se, Mo, Co, Sr, Pb, Pt, Sb, Y	Macro: 12.13–2774 Micro: 0.04–99.89	[1]
*N:* Cocoa	Macro: Na, Ba, Fe, Mg, Al Micro: Cr, La, Ce, Mo, Cs, Ga, Ti, Y, Cu, Cd, Mn, Ni, As, Pb, V, Co, Rb, Zn, Sr,	Macro: 5.4–3200 Micro: 0.01–25	[41]
*N:* Peanuts *(Arachis hypogaea* L.*)*	Macro: ^11^B, ^23^Na, ^54^Fe, ^56^Fe Micro: ^59^Co, ^52^Cr, ^53^Cr, ^58^Ni, ^60^Ni, ^61^Ni, ^62^Ni, ^82^Se, ^63^Cu, ^65^Cu, ^98^Mo, ^66^Zn	Macro: 16.9–7440 Micro: 0.0206–28.3	[42]
*F:* Tomato (*Lycopersicon esculentum*)	Macro: Al, Ba, Fe, Micro: Cd, Co, Cr, Cu, Hg, Mn, Ni, Se, Sn, Sr, V, Zn	Macro: <0.025–57.8 Micro: <0.008–29.5	[43]
*N:* Wheat, millet, corn, soybean	Macro: Li, K, Mg, Ca, Fe Micro: Zn, Mn, Cu, Mo, Se	Macro: 0.012–2285 Micro: 0.0–15.11	[44]
*N:* Rice	Macro: ^23^Na, ^25^Mg, ^44^Ca, ^56^Fe, Micro: ^65^Cu, ^66^Zn, ^75^As, ^111^Cd, ^55^Mn,	Macro: 0.049–28 Micro: 0.02–9.4	[45]
*N:* Cocoa	Micro: Cd, Sb, Pb, As, Cr, Se, V	0.026–1.315	[46]
*F:* Passion *(Passiflora edulis)*, star anise *(Illicium verum)*	Micro: ^75^As, ^112^Cd, ^59^Co, ^53^Cr, ^65^Cu, ^55^Mn, ^60^Ni, ^208^Pb, ^88^Sr, ^51^V, ^64^Zn	0.034–276	[47]
*F:* Cape gooseberry *(Physalis peruviana* L.*)*	Micro: Co, Cu, Se	0.03–6.7	[48]
*V:* Tea	Macro: Fe, K, Br, Ca Micro: Cu, Cr, Si	Macro: 86–6785 Micro: 2–124	[49]
*T: Alpinia oxyphylla* and *Morinda officinalis*	Macro: Mg, K, Ca, Na, Fe, Al, Ba, Micro: Zn, Mn, Cu, Mo, Cr, Ni, As, Se, Cd, Hg, Tl, Pb, V	Macro: 26.18–24,890 Micro: 0.66–186	[50]
*F: Capsicum annuum, C. chinense* and *C. frutescens*	Macro: Mg, P, K, Ca, B, Fe, Micro: Cu, Zn Mn,	Macro: 30–75,600 Micro: 2–185	[51]
*N:* Pea (*Pisum sativum* L., *cv. Bohun)*	Micro: Cd, Cu, Pb, Zn	3.3–425	[52]
*F:* Black, green olive	Macro: Mg, Fe Micro: Cr, Co, Ni, Cu, Zn, Sn, Cd, Pb	Macro: 7.08–79.28 Micro: 0.06–39.06	[53]
*F:* Blueberry, Strawberry	Macro: ^11^B, ^23^Na, ^24^Mg, ^27^Al, ^31^P, ^39^K, ^43^Ca, ^57^Fe Micro: ^53^Cr, ^55^Mn, ^60^Ni, ^63^Cu, ^66^Zn, ^72^Ge, ^75^As, ^82^Se, ^111^Cd, ^208^Pb	Macro: 1.54–5960 Micro: 0.0057–142	[54]
*N:* Macadamia, lotus, pistachios, sunflower, pine, almond, walnut, chestnut, hazelnut, cashew, ginkgo	Macro: Li, Ba Micro: Cr, Mn, Co, Cu, Zn, As, Se, Rb, Sr, Mo, Cd, Cs, Pb, Th, U	Macro: 0.01–4.5 Micro: 0.0004–40	[55]
*N:* Rice (*Oryza sativa* L.)	Micro: Zn, Cd, Sb, Pb, As	N.A	[56]
*V:* Safflower (*Carthamus tinctorius* L.)	Macro: Al, Ca, Fe, Mg Micro: As, Cd, Cu, Hg, Pb, Co, Cr, Mn, Mo, Ni, P, Se, Sr, V, Zn	Macro: 565–12,484 Micro: 0.046–108.43	[57]
*V:* Sunflower (*Helianthus annuus*)	Macro: Ca, Fe, K, P, S Micro: Cd, Ce, Cr, Cu, La, Mn, Ni, Zn	Macro: 1000–35,800 Micro: 1.01–80	[58]
*N:* Rice, wheat *T:* Panax *V:* Celery, astragalus, spirulina, garlic	Macro: K, Na, Ba, Ca, Mg, P, Micro: ^59^Co, ^133^Cs, ^63^Cu, ^95^Mo, ^60^Ni, ^208^Pb, ^85^Rb, ^232^Th, ^238^U, ^45^Sc, ^89^Y, ^140^Ce, ^163^Dy, ^166^Er, ^153^Eu, ^157^Gd, ^165^Ho, ^139^La, ^175^Lu, ^146^Nd, ^141^Pr, ^147^Sm, ^159^Tb, ^169^Tm, ^172^Yb, ^9^Be, ^209^Bi,	Macro: *N:* 50–250 *T:* 44–1050 *V:* 5.8–1300 Micro: *N:* 0.0002–0.85 *T:* 0.00028–0.36 *V:* 0.00012.4–0.213	[59]
*N:* Soybean	Micro: Cu, Zn	N.A	[60]
*N:* Rice in southern Brazil	Macro: Fe, Mg Micro: As, Cd, Co, Cu, Mn, Ni, Se, Zn	Macro: 15.56–94 Micro: 0.00045–0.5	[61]
*V:* Tea	Macro: Fe, Al Micro: Pb, As, Cd, Cu, Zn, Se Mo, Cr	Macro: 1510–3100 Micro: 0.034–197	[62]
*F:* Pepper, tomato, aubergine, apricot, peach, plum, olive, grape, prune, zucchini *N:* Pea, hazelnut, walnut *T:* Potato, turnip *V:* Lettuce, chicory, endive, cabbage, fennel	Micro*:* Co, Cr, Cu, Mn, Mo, Ni, Sr, Tl, U, V, Zn	*F:* 3 × 10^−4–^ 5.5 *N:* 2 × 10^−4^–21 *T:* 9 × 10^−4^–5.34 *V:* 1,2 × 10^−4^–3.82	[63]
*F:* Tomato *V:* Artichoke	Micro: ^75^As, ^60^Ni, ^202^Hg, ^51^V, ^208^Pb, ^111^Cd, ^63^Cu, ^52^Cr	*F:* 3 × 10^−6^–1.934 *V:* 3 × 10^−6^–2.378	[64]
*N:* Maize *T:* Potato, radish *V:* Cabbage, pakchoi, scallion, garlic, lettuce, parsley, tine peas, spinach	Micro: Cd, Cr, Cu, Pb, Zn	*N:* 0.8–12.9 *T:* 1.1–64.8 *V:* 0.3–90.4	[65]
*F:* Hot pepper *N:* Rice, bean *T:* Carrot, radish, potato *V:* Cabbage, *H. houttuyniae*, celery, garlic stem	Micro*:* Hg, Pb, Cd, Mn, Se	*F:* 0.00013–0.0426 *N:* 0.00001–0.0308 *T:* 0.00004–0.0122 *V:* 0.00102–0.646	[66]
*F:* Banana, mango *N:* Maize *T:* Cassava *V: Amaranthus tricolour*	Micro: Zn, Cu, Co, Ni, As, Cd, Cr, Pb, Mn	*F:* 0.0–7.14 *N:* 0.0–16.3 *T:* 0.0–15.4 *V:* 0.0–25	[67]

Note: *F*: Fruit; *N*: Nut; *T*: Tuber; *V*: Vegetable.

This systematic review aims to gather research on the application of ICP-MS combined with PCA as a technique to establish the authenticity of the geographical origin of plant-based foods, which could be considered to promulgate as an international standard. The review focuses on unprocessed plant-based foods such as cereals, rice, wheat, maize, sorghum, ragi, pulses, legumes, fruits, vegetables, and nuts. Origin fraud, which occurs when plant food is misrepresented in its geographical origin, is a form of mislabeling that has a significant impact on the economy and is documented in many countries. Agricultural and food products are subject to strict control on their origin to ensure quality during import and export [68,69,70]. Currently, different types of standards exist for agricultural and food products [71,72], but standards for determining the geographical origin have not yet been widely adopted.

Several common multivariate statistical methods are used for geographical origin determination, including principal component analysis (PCA), cluster analysis (CA), linear discriminant analysis (LDA), canonical discriminant analysis (CDA), and hierarchical cluster analysis (HCA). Among these, PCA is the most commonly used method due to its ability to effectively distinguish data with similar characteristics [73]. Therefore, this paper specifically focuses on the use of PCA for plant food traceability.

## 2. Methods

### 2.1. Multielement and Accuracy Analysis

#### 2.1.1. Multielement Analysis

Generally, trace elements represent the geographical tracer in a specific soil condition, and are absorbed via the roots and transferred to various parts of the plant. The distribution of trace elements reflects the elemental signature of the soil origin. In addition, the isotope ratios of the elements show the linkage between products and soil characteristics. Particularly, isotopes of heavy metals have been considered the most suitable for tracing a plant-based food’s origin. However, the isotopes of light elements such as hydrogen, nitrogen, oxygen, and sulfur are considered reliable indicators of food authentication, but the ratio of these elements is too variable to serve as tracers of the soil where a product is produced [74,75,76].

ICP-MS is a robust analytical technique for the determination of multi-elemental composition (qualitatively), concentration (quantitatively), and isotopic abundances of various matrices. The structure and operating principles of the ICP-MS device have been presented in numerous previous documents. The ICP-MS analyzer can detect many elements and simultaneously identify their isotopes.

Out of the 118 elements in the periodic table, 16 elements are not recommended for measurement on ICP-MS and 44 elements cannot be measured on the ICP-MS instrument.

The elements not recommended for ICP-MS are B, Si, Cl, Ca, Br, Hg, P, S, Zr, Nb, Tc, I, Hf, Ta, W, and Os.The elements that cannot measured by ICP-MS are H, He, C, N, O, F, Ne, Ar, Kr, Xe, Pm, Po, At, Rn, Fr, Ra, Ac, Pa, Np, Pu, Am, Cm, Bk, Cf, Es, Fm, Md, No, Lr, Rf, Db, Sg, Bh, Hs, Mt, Ds, Rg, Cn, Nh, Fl, Mc, Lv, Ts, and Og.

Out of the remaining 58 elements, rare earth elements (REEs) can act as geochemical markers, however, less information using REEs in foodstuff traceability [77]. Additionally, there are other elements with low content in food samples, such as Ga, Ge, Rb, Y, Ru, Ir, Au, U, and Te. After removing the elements that are not present, 36 elements are left in the food sample. The elements selected for analysis on ICP-MS are presented in Table 2. These 36 elements are commonly analyzed using ICP-MS methods for elemental or multi-elemental determination in food traceability.

#### 2.1.2. Sample Preparation

Solid samples are digested in strong and hot acid conditions, such as HNO_3_, HNO_3_/HCl, HNO_3_/H_2_O_2_, or HNO_3_/HF, which depend on the specific matrices. In general, samples are commonly digested with pure HNO_3_ (65–70%) in a microwave oven, and then diluted with ultra-pure water [78]. There are various methods to convert solid samples into aerosols, including electrothermal vaporization (ETV), laser ablation (LA), microwave-assisted digestion (MAD), spark ablation, etc. The samples are then transported to the plasma by an inert gas. In these techniques, the ETV analysis method is used for combustible samples while the spark ablation is applied for conducting samples in sampling large spots with a diameter of 1–3 mm. The LA microanalysis technique uses high-irradiance (UV) lasers to measure very small spots (2–750 µm in diameter) on almost solid samples whilst the MAD method is applied for the sample preparation process in the analysis by ICP-MS, inductively coupled plasma atomic emission spectrometry (ICP-AES), graphite furnace atomic absorption spectrophotometry (GFAA), and flame atomic absorption spectrophotometry (FLAA).

### 2.2. PCA Tools

PCA is a popular multivariate statistical algorithm program to distinguish components from each other via six main steps by transforming a vector into a matrix in mathematics [79]. PCA plays an important role in reducing the dimensionality of complex datasets, changing them to a more simple and easier status, and minimizing information loss [80]. PCA and LDA are considered as the most powerful discriminators of data in multivariate analysis tools, which are commonly used to discriminate the geographical origin of plant-based agricultural products. While both PCA and LDA techniques were applied to identify linear combinations of features in the best data explanation, LDA is a technique reducing the supervised dimension that achieves the simultaneous data classification. LDA concentrates on finding a feature subspace which helps to enhance the separability between groups, whilst PCA is an unsupervised technique that disregards class labels and concentrates to capture the maximum variation direction in the datasets [79,80].

Figure 1 shows the different number of research publications using PCA, LDA, K-nearest neighbor (KNN), and HCA multivariate statistical methods for geographical origin determination. PCA is a popular technique used in determining the geographical origin of agricultural products because it can reduce dimensionality by using main principal components (PCs) to express the information spread across numerous columns, wherein the first few PCs can account for an important proportion of the total variance. These PCs are then used as explicable variables in machine-learning models. In addition, datasets with more than three features or dimensions can be difficult in class visualization. It can be observed that a clear distinction between clusters or classes relies on the first two PCs, which allows for a simple and more effective visualization of the data [81].

Principal component analysis possesses some advantages: it is an effective computation algorithm that can speed up machine learning processes and prevent data overflow; PCA can improve the performance of machine learning (ML) algorithms by eliminating unnecessary correlated variables; the variance of the PCA is high, which allows better visualization of the data; and PCA can contribute to reducing noise, which cannot be automatically ignored, making it a valuable tool for data analysis. On the other hand, a few disadvantages of principal component analysis have been reported, including that PCA can sometimes be difficult to interpret, particularly when identifying the most necessary characteristics even after the calculation of the major components; the calculation of covariances and covariance matrices may sometimes be challenging; and, in some particular cases, the computed principal components might be more difficult to understand than the original set of components.

Principal Component Analysis (PCA) can be a complex statistical technique that requires expertise in mathematics. However, there are several software programs available that make it easier for non-specialists to perform PCA. These programs can be particularly useful for determining the geographical origin of a sample. To this end, Table 3 lists several commonly used software options.

In addition to the widely used software packages mentioned earlier, there are many other specialized software programs built for specific purposes depending on the type of data being analyzed. Some of these specialized software packages are designed for genomics, proteomics, medical imaging, or other specialized fields. The availability of these specialized software packages reflects the diverse and complex nature of modern data analysis.

While most software packages use similar algorithms and produce similar results, each package has its unique features, interfaces, and outputs. Some software packages, such as XLSTAT, have a vivid visual interface that makes it easy for non-expert users to interact with the software and interpret results. Other software packages, such as R, offer more advanced customization options and allow for greater flexibility in data analysis.

The ability to customize the analysis and interpret the results accurately is critical in scientific research and decision-making. Therefore, the choice of software package depends on the specific research question, the type of data being analyzed, and the expertise of the user. In addition to the analytical capabilities, software packages also offer various chart and graph options to visualize the data in different ways, allowing users to communicate the results more effectively. Overall, choosing the right software package is essential for efficient and effective data analysis.

There are numerous specialized software programs available for various purposes, depending on the type of data being analyzed. Most software requires a license to be purchased. While the software generally produces similar results due to using similar algorithms, each program offers different information, features, and interfaces. Among the various software programs, XLSTAT is often preferred due to its graphical representation and user-friendly interface. However, professional users tend to utilize R software, which allows for greater flexibility in tweaking the code to produce more accurate data (Figure 2).

## 3. Discussion

### 3.1. Applications of ICP-MS Combined with PCA for Determining the Origin of Agricultural Products

In recent years, numerous scientific studies worldwide have been conducted to successfully develop methods for determining the origin of food products for different agricultural commodities. Many of these studies have utilized the ICP-MS analysis method combined with PCA to determine the origin of various food products, such as wine [90,91], pork [7], sheep [92], mutton [15], bivalve mollusks [93], and sea cucumbers [94]. Meanwhile, studies on tracing the origin of plant-based food products are summarized in the Table 4.

The Table 4 illustrates that a large number of samples are necessary to determine the origin of food products. Typically, more than three samples are required from a single region. The greater the number of samples collected, the more accurately the distinctive characteristics of the region can be identified, resulting in a more precise identification. Additionally, a greater number of elements need to be identified using ICP-MS than in other studies, with a minimum of 20 elements being the most effective. Fewer elements lead to less information for identification and inaccurate results. Conversely, if there is insufficient data, the use of PCA statistics will not produce complete information. This is similar to having high precision but low accuracy. These factors highlight the significant effort, time, and analytical costs involved in food traceability studies, as more information is required than in other types of research [132,133].

Table 4 reveals that rice is the most extensively researched commodity worldwide in terms of traceability. The authenticity of rice has increasingly become a crucial issue in recent years. To authenticate rice, a range of techniques has been employed, such as determining its geographical origin, distinguishing between different cultivars, verifying organic rice authenticity, and detecting impurities in rice [134].

### 3.2. The Necessity of Standardization for Geographical Traceability Methods

#### 3.2.1. Current Related Standard

Although various traceability studies have been conducted on different food products as presented above, the current method of food traceability is still not regulated, making it difficult to accurately evaluate. Quality standards and packaging regulations are primarily set to evaluate the quality of products and determine their origin [80]. Several compelling factors are driving the need for accurate analytical methods to authenticate the origin of our food. The UK Food Standards Agency (FSA) has solicited public input on various food labeling issues. According to the FSA’s research, “country of origin labeling” ranked high on consumers’ list of demands for change [135].

Geographical indications (GIs) refer to the use of place names to identify products that originate from specific regions and protect their quality and reputation. They are commonly used for wines, spirits, and agricultural products. By granting certain foods recognition for their distinctiveness, GIs differentiate them from other foods in the marketplace, making them commercially valuable. GIs may also provide relief from acts of infringement or unfair competition and protect consumers from deceptive or misleading labels. Some examples of registered or established GIs include Parmigiano Reggiano cheese and Prosciutto di Parma ham from the Parma region of Italy, Toscano olive oil from Tuscany, Roquefort cheese, Champagne from the region of the same name in France, Irish Whiskey, Darjeeling tea, Florida oranges, Idaho potatoes, Vidalia onions, Washington State apples, and Napa Valley Wines. Over the past decade, determining the geographical origin of food has become an increasingly important issue for countries worldwide. Consumers are concerned about the authenticity of the food they eat [75].

To carry out geographical indications, scientists need to conduct a series of studies on chemical composition analysis using various methods of determination. This is a complex and expensive task that can sometimes be too costly for businesses to afford [61,99,136]. On the other hand, without experience, one would have to search for suitable methods, which would take a lot of time and effort because there is no pre-defined method. Therefore, issuing a feasible international standard method that can distinguish the geographical origin of plant foods will help reduce costs and time in PGI research. This will improve efficiency in agricultural production.

#### 3.2.2. Main Steps of the Proposed Standard Method

Undoubtedly, creating standards for determining the origin of plant-based food is of the utmost importance. Based on the data collected, several essential steps need to be incorporated into the development of the standard, as illustrated in Figure 3. This entails the integration of two processes: (1) the profiling method and (2) the geographical traceability method. The profiling method involves the following key steps:

*Step 1—Sample Collection*: the received samples must ensure relevant information about the variable, geographical details, and coordinates. The collection of samples should include at least 5 or 10 samples per geographical region.*Step 2—Sample Analysis* includes two methods: sample preparation and analysis. These methods are built depending on the equipment of each laboratory. However, it is necessary to ensure the analysis of at least 20 elements on the ICP-MS equipment. When building the standard, it is necessary to specify which elements and parameters of the method are included.*Step 3—Input data into PCA:* it is necessary to set parameters for the PCA software. The PCA method needs to determine accuracy and reliability.

In the initial stages of the geographical traceability method, the sample of interest is an unknown entity. These samples are taken to the laboratory to undergo meticulous analysis using ICP-MS. The resulting multielement data is then fed into the PCA, allowing for effective differentiation and source identification. It is imperative that the findings are presented with precision and reliability.

## 4. Conclusions

The present review summarizes the research on the application of ICP-MS and PCA in the geographical origin authentication of agricultural products. Consequently, ICP-MS is a robust, accurate, and highly sensitive technique for determining the inorganic elements in food substances, whereas PCA can reduce dimensions, speed up machine learning processes, prevent data overflow and reduce noise. The combination of ICP-MS and PCA can be considered a powerful tool and a standardized approach to authenticating and certificating the geographical origin of plant-based foods, which plays an important role in protecting quality products. In addition, this might be the base for producers making decisions to enhance the effectiveness of the certification of their products to match the demand of consumers in the markets.

## Figures and Tables

**Figure 1 foods-12-01848-f001:**
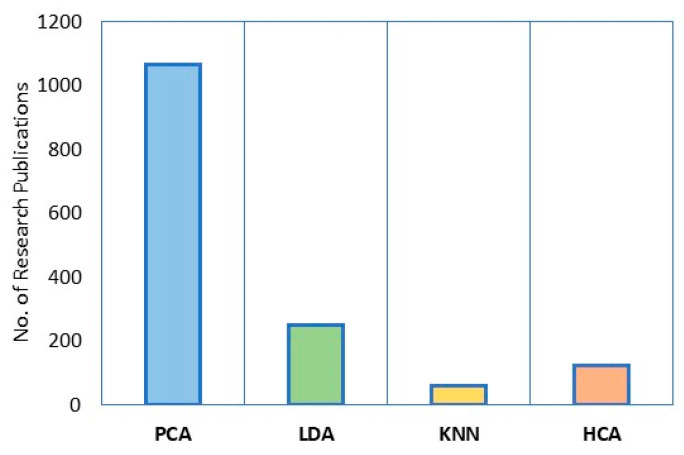
Number of research publications using a multivariate algorithm to discriminate geographical origins.

**Figure 2 foods-12-01848-f002:**
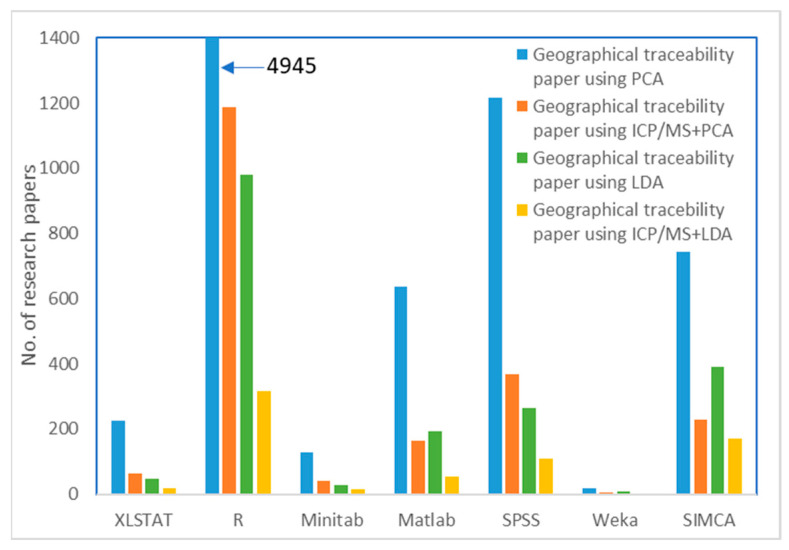
Research publications using ICP-MS and PCA/LDA to discriminate geographical origins. (The data was summarized and collected from various research, which was published in the ScienceDirect system).

**Figure 3 foods-12-01848-f003:**
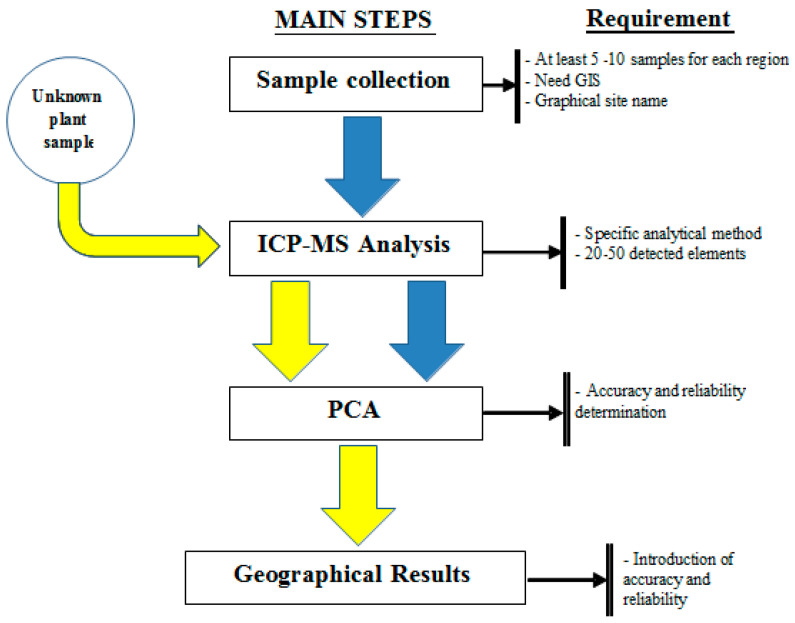
Proposed standard method.

**Table 2 foods-12-01848-t002:** 36 selected metals for common analysis by ICP-MS.

* ^mass^ * ** *Element* **	^7^ **Li**	^9^ **Be**	^23^ **Na**	^24^ **Mg**	^27^ **Al**	^28^ **Si**	^39^ **K**	^40^ **Ca**	^45^ **Sc**
** *Abundance (%)* **	92.4	100	100	78.99	100	92.23	93.26	96.94	100
* ^mass^ * ** *Element* **	^48^ **Ti**	^51^ **V**	^52^ **Cr**	^55^ **Mn**	^56^ **Fe**	^58^ **Ni**	^59^ **Co**	^63^ **Cu**	^64^ **Zn**
** *Abundance (%)* **	73.72	99.75	83.79	100	91.75	68.08	100	69.17	48.63
* ^mass^ * ** *Element* **	^75^ **As**	^80^ **Se**	^88^ **Sr**	^95^ **Mo**	^105^ **Pd**	^107^ **Ag**	^111^ **Cd**	^118^ **Sn**	^121^ **Sb**
** *Abundance (%)* **	100	49.61	82.58	15.92	22.33	51.84	12.80	24.22	57.21
* ^mass^ * ** *Element* **	^138^ **Ba**	^182^ **W**	^185^ **Re**	^195^ **Pt**	^201^ **Hg**	^205^ **Tl**	^208^ **Pb**	^209^ **Bi**	^232^ **Th**
** *Abundance (%)* **	71.70	26.50	37.40	33.83	13.18	70.48	52.4	100	100

**Table 3 foods-12-01848-t003:** Software most commonly used for geographical origin determination.

Software	Description	Feature	Ref
**XLSTAT** Version 2023 5.1.1407.0 United States	A popular data analysis add-on for Microsoft Excel, known for its flexibility and powerful statistical tools. It offers a wide range of features and functions, including PCA, which can be used for geographical origin determination. With XLSTAT, users can easily analyze, customize, and share their results within the familiar Microsoft Excel interface.	User-friendly interface that makes it easy for non-experts to use. Some of its features include descriptive statistics, hypothesis testing, ANOVA, regression analysis, time series analysis, data visualization, and machine learning. It also offers advanced statistical tools such as principal component analysis (PCA), partial least squares regression (PLS), and discriminant analysis. XLSTAT offers a wide range of customizable and interactive charts, including scatter plots, line charts, bar charts, histograms, box plots, and more. The charts are designed to be easy to interpret and can be modified to fit specific needs.	[82]
**R** 4.3.0 France	A free, open-source programming language for statistical computing and graphics which was developed at Bell Laboratories. It is widely used by researchers and analysts in various fields such as economics, finance, biology, and social sciences. R provides a wide range of statistical and graphical techniques, including linear and nonlinear modeling, statistical tests, time-series analysis, and data visualization. Additionally, R has a large and active community of users who contribute to the development of packages and resources that extend the functionality of the language. Due to its flexibility, power, and cost-effectiveness, R has become a popular choice for data analysis and research.	PCA uses functions such as princomp and prcomp. While princomp uses covariance matrix decomposition, prcomp uses singular value decomposition, which often provides better numerical accuracy. Moreover, R offers a vast array of packages that can be used to implement PCA, such as ade4, vegan, ExPosition, dimRed, and FactoMineR, among others. These packages provide additional functionalities beyond the basic PCA function, such as biplots, scree plots, and visualization of the results, which are useful for interpreting and communicating the analysis outcomes. Additionally, R has a user-friendly interface and a large community of users who share their scripts and offer support for beginners.	[83]
**Minitab** 21.1.0 United States	A statistical software package developed for quality improvement and statistical analysis. It provides a wide range of tools for data analysis, including graphical tools, statistical tests, and regression analysis. Minitab also supports statistical process control, the design of experiments, and Six Sigma methodologies. With its user-friendly interface and extensive documentation, Minitab is a popular choice for quality professionals and data analysts.	A wide range of features, including statistical analysis, data visualization, and predictive analytics. It offers a user-friendly interface with easy-to-use tools for data analysis and quality improvement. Minitab also has built-in templates for various industries, including manufacturing, healthcare, and finance. Some of its popular features include descriptive statistics, hypothesis testing, regression analysis, and design of experiments. Additionally, Minitab can be used to create interactive graphs and charts to help visualize and present data in a clear and concise manner. Minitab has some limitations, such as: - Limited support for programming and scripting compared to other statistical software. - Limited graphics customization compared to other software. - Limited support for advanced statistical analyses, such as structural equation modeling and item response theory.	[84]
**Matlab** R2022a United States	A high-performance language for technical computing and data analysis. It is widely used in academia, industry, and research for developing algorithms, analyzing and visualizing data, and building models. Matlab is one of MathWorks products.	Matlab has several functions and tools for performing PCA, such as princomp and PCA. These functions can perform PCA on matrices, including missing data and scaled variables. Matlab also allows for the customization of PCA outputs and visualization of the results. Other features of Matlab for PCA analysis include: - Ability to handle large datasets and perform PCA on high-dimensional data - Capability to extract and plot biplots, which display the relationships between variables and observations in a single plot - Tools for assessing the significance of principal components and identifying outliers - The option to perform PCA on specific subsets of data, such as specific rows or columns of a matrix - The ability to save PCA results and load them for future analysis or comparison.	[85]
**SPSS** Statistics Version 26 United States	SPSS (Statistical Package for the Social Sciences) is, a IBM product, widely used statistical software in various fields, such as psychology, marketing, healthcare, and education. It has a broad range of statistical analysis options, making it a versatile tool for data analysis. Additionally, SPSS allows for data cleaning, data transformation, and data management, which are essential steps in the data analysis process. With SPSS, users can conduct various multivariate techniques, including principal component analysis, factor analysis, cluster analysis, and discriminant analysis, among others. The software is regularly updated to incorporate the latest statistical techniques and methods, making it a reliable and up-to-date tool for data analysis.	SPSS offers a variety of features for data analysis, including statistical analysis, data mining, and predictive analytics. Some of its key features related to PCA include the ability to perform principal component analysis to identify underlying structure in the data and reduce its dimensionality, the option to perform factor analysis to identify latent variables underlying observed variables, and the ability to generate graphical output to visualize the results. SPSS also offers a user-friendly interface, making it accessible to non-technical users, as well as a range of advanced statistical techniques for more experienced users. Additionally, SPSS allows users to automate analysis and reporting, making it a time-efficient option for large datasets.	[86,87]
**Weka** 3.8.3 New Zealand	A collection of machine learning algorithms for data mining tasks was developed at University of Waikato. It is open-source software written in Java. Weka includes tools for data pre-processing, classification, regression, clustering, association rules, and visualization. It provides a graphical user interface for exploring data and running machine learning algorithms, as well as a command-line interface for batch processing and integration with other software systems. Weka is widely used in both academia and industry for research, education, and practical applications in areas such as bioinformatics, text mining, image analysis, and more.	Weka has a built-in PCA algorithm that can perform principal component analysis on data sets. It allows users to select the number of principal components to be extracted and provides options for normalization and centering of the data. Weka also provides visualizations of the data, including scatter plots and parallel coordinate plots, to aid in understanding the results of the PCA analysis. Additionally, Weka’s PCA algorithm can be used in combination with other machine learning algorithms available in the software for tasks such as classification and clustering. Weka provides several visualization tools for exploring and interpreting the results of PCA, including scatter plots, biplots, and correlation matrices. These charts can help users to better understand the relationships between variables and identify patterns in the data. Additionally, Weka supports the export of charts and graphs to various formats, such as PNG and PDF, for easy sharing and presentation of results.	[88]
**SIMCA** 17.0 Germany	SIMCA (Soft Independent Modeling of Class Analogy) is a multivariate data analysis software developed by Umetrics AB that is widely used for classification and predictive modeling in various industries. It uses PCA to reduce the dimensionality of data and identify relevant variables for modeling. SIMCA is particularly popular in the pharmaceutical, chemical, and food industries for quality control and process optimization purposes.	SIMCA is a powerful tool for multivariate data analysis, including PCA. It offers a user-friendly interface and intuitive data visualization tools, such as scatter plots and score plots, to help users understand their data. Additionally, SIMCA has a range of advanced features for outlier detection and model validation. These capabilities make it a valuable tool for analyzing large datasets and identifying patterns in complex data structures.	[89]

Note: the websites of software are listed corresponding to: XLSTAT (xlstat.com), R (r-project.org), Minitab (minitab.com), Matlab (mathworks.com), SPSS (ibm.com), Weka (cs.waikato.ac.nz), SIMCA (sartorius.com). Accessed on 27 April 2023.

**Table 4 foods-12-01848-t004:** ICP-MS and PCA in plant food traceability studies.

Plant	Region (Country)	No. Sample/Region	No. Element	Ref.
** *Fruit type:* **				
Cherry	Regina, Kordia, Mpakirtzeika, Skeena (Greece)	78/4	25	[95]
	Ferrovia, Canada Giant, Lapins, Germersdorfer (Edessa and neighbouring Kozani region, Greece)	56/4	25	[96]
Jackfruits	North 24 Parganas, Nadia, West Tripura, Khowai, Panruti, Varkala, South Sikkim (India)	70/7	24	[97]
	Nadia, North 24 Parganas and South Sikkim (India)	70/3	24	[98]
Lemon	Tucumán, Jujuy, Corrientes (Argentina)	74/3	25	[99]
	Sicily (Italy), Çukurova (Turkey)	40/2	32	[100]
Pear	10 locations in Fundão (Portugal)	150/10	24	[101]
	Italy, Spain, Greece, Cyprus	74/4	19	[102]
** *Nut type:* **				
Almonds	Australia, Italy, Iran, Morocco, Spain, United States of America	250/6	58	[58]
	Sicily, Spain and California	21/3	7	[103]
Maize	Jilin, Gansu, Shandong (China)	90/3	25	[104]
*Ricinus communis*	Brisbane, Far North Qld, West Qld, South Sydney, West Sydney, Newcastle, North Coast NSW, North Adelaide, South Adelaide, East Adelaide, South Coast Adelaide, North Perth, East Perth, Fremantle, Inner East Melbourne, West Melbourne, East Melbourne, Swan Hill (Australia)	68/18	92	[105]
Rice	Jiansanjiang, Wuchang, Chahayang (Heilongjiang, China)	237/3	33	[106]
	Fuzhou, Longyan, Nanping, Ningde, Putian, Quanzhou, Sanming, Xiamen, Zhangzhou (China)	206/9	13	[107]
	Goiás, Rio Grande do Sul (Brazil)	31/2	20	[108]
	Italy, Turkey	40/2	21	[109]
	Suwon (Korea), Shanghai (China), Los Banos (Philippines)	27/3	25	[42]
	Heilongjiang, Jinlin, Zhejiang, Jiangsu, Hunan, Guizhou (China)	39/6	25	[110]
	Campanha, Central, Fronteira Oeste, Planície Costeira Interna, Sul	640/5	26	[111]
	Fengshan, Donglan, Bama, Rugao, Yangdong, Jiaoling, Sanshui, Huaiji, Guangning, Sihui, Songtao, Qianxi, Fuquan, Tongren, Kaili, Guang’an, Nanchong, Mianyang, Chengdu, Luzhou, Changshou, Tieling, Dandong, Suihua, Baicheng, Huinan, Xinxiang, Xinyang (China)	84/28	27	[112]
	Wuchang, Qiqihar, Jiamusi (China)	194/3	16	[57]
	Cambodia, Japan, Korea, Philippines, Thailand	59/5	29	[113]
	Anhui, Guangxi, Guangdong, Jilin, Heilongjiang, Inner Mongolia (China)	18/6	30	[114]
	Gujranwala, Gujrat, Narowal, Wazirabad, Chiniot, Okara, Bahawalpur, Bahawalnagar, Faisalabad, Sahiwal, Jhang, Lodhran (Pakistan)	64/12	35	[115]
	Jansanjiang, Wuchang, Chahayang (China)	92/3	52	[116]
Sesame	Gondar, Humera, Wollega (Ethiopia)	93/3	12	[117]
	Korean, Chỉnese and Indian	123/3	15	[118]
Soybean	Bei’an, Nenjiang, Heihe, Heilongjiang (China)	42/4	24	[119]
	Ha Giang, Hanoi, Dong Nai (Vietnam); Ontario, Manitoba (Canada); Iowa, Illinois (United States); Mato, Grosso (Brazil)	38/9	40	[120]
Wheat	Hebei, Henan (China)	61/2	22	[121]
	Hebei, Henan, and Shanxi provinces (China)	270/3	13	[122]
** *Tuber type:* **				
Potato (grown)	Alpine, Dinaric, Mediterranean, Pannonian (Slovenia)	36/4	25	[123]
	Abruzzo, Lazio, Molise, Puglia, Emilia Romagna, and Veneto (Italia)	30/6	10	[124]
** *Vegetable type:* **				
*Asparagus*	Poland, Greece, Spain, Peru, China, Germany, Netherlands	319/7	36	[125]
	Shandong, Hebei, Lianing (China)	22/3	15	[126]
Cabbages	Dandong, Yantai, Zhangjiakou, Qingzhou, Pingdu, Hangzhou, Shanghai (China); Gyeonggi, North Chungcheong, South Chungcheong, Gangwon, North Gyeongsang, South Gyeongsang, South Jeonla (Korea)	363/14	22	[127]
	Gangwon, North Gyeongsang, South Gyeongsang, South Jeonla, and South Chungcheong (Korea) and Qingzhou, Pingdu, Yantai, Dandong, and Zhangjiakou (China)	160/10	19	[128]
Mushroom	Chu Xiong, Da Li, Yu Xi (China)	40/3	13	[129]
	Bulgaria, Romania, Croatia, Hungary, Iran, Slovenia, Italy, Spain, Australia, and China	64/10	45	[130]
Pakchoi	Tien Phong, Thanh Da, Linh Nam, Thanh Xuan, Van Duc, Van Noi (Vietnam)	60/6	42	[131]

## Data Availability

Not applicable.

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
