# Peer review of "Towards a Standardized Approach for the Geographical Traceability of Plant Foods Using Inductively Coupled Plasma Mass Spectrometry (ICP-MS) and Principal Component Analysis (PCA)"

_foods, 2023, doi:10.3390/foods12091848_

Round 1
Reviewer 1 Report
GENERAL COMMENTS
The manuscript consists an effort to review most of research about using ICP-MS combined with the PCA for the geographical origin determination of plant-based products, and simultaneously convince their readers of the necessity this method to be standardized. The review covers some of the goals but some revisions are needed.
It would be better if the authors reorganize the structure of the article clearly showing the discussion section. Please, consider sections 2 and 3 to be under one section of Methods, while sections 4 and 5 to be the Discussion, for instance.
It is recommended that authors should enhance and further explain their statements by giving and discussing few examples from the literature. This is a part which is missing from the review.
Please, add references to the manuscript and check out the grammar and syntax of the manuscript.
Please, check out the reference part. Not all of them are written in the appropriate type for Foods Journal.
TITLE
Line 1: Please replace “Communication” with “Review”.
Line 3: I would suggest to replace the slash (/) in the abbreviation of ICP/MS with a hyphen (-) which is the most appropriate. Please correct that in all cases throughout the manuscript. It is correctly written in lines 73, 80, 82. Consider to add the full name of ICP-MS before its abbreviation.
Line 3: Consider to add the abbreviation in brackets of (PCA) after the full name.
Line 4: Correct the superscripts in the first name (*12 to 1,2,*)
ABSTRACT
Line 11: Since ICP-MS and PCA are referred for the first time to the text, the full names should be written and the abbreviations should be in brackets.
Line 13: Please, check out the grammar of this sentence.
Lines 19-21: “Furthermore, the paper proposes the main steps needed to establish the standard for this traceability method, including the development of 20 guidelines and quality control measures.” Please, confirm if the authors are able to propose in a systematic review. The main purpose of a systematic review is to summarize all the literature research in responding to a specific research question. The answer should be extracted and supported by the studies of the literature.
1. INTRODUCTION
Lines 35-37 (+Table 1): Are those studies that investigate multi-elemental analysis in plant-based foods using PCA statistical method? Please, clarify.
Lines 54-57: Please, check out the grammar of this sentence.
Line 55: Please, correct “…geographical origin…”.
Line 63: Please, add a reference: “Currently, different types of 62 standards exist for agricultural and food products [Ref.], but standards for determining the geographical origin have not yet been widely adopted.”
Lines 65-70: Please, add references.
2. MULTIELEMENTAL AND ACCURACY ANALYSIS
This section is suggested to be rewritten. It is a section which needs major revisions. Please, try to be more precise and clarifying on both sub-sections (Multielement analysis, Sample preparation). Please, add references to this section in order to support your statements.
In “Multielement analysis” sub-section, a brief explanation of why there is a need to investigate the elements for geographical discrimination, why and when are we interested in elements and their isotopes, why are there elements are not recommended and why are there elements that cannot be measured.
Line 80: Please, correct the following “The elements are not recommended…”
Line 82: Please, correct the following “The elements cannot be measured …”
Line 85: Please, try to keep a consistent Table 2 (make all elements in Bold etc.). Please, move the table after the last paragraph.
Line 86-87: The following statement is not true: “...there are elements of the Lanthanum family that 86 are not present in food samples...”. This information is misleading. There are studies which investigate the present of Lanthanides in plant-based foods (i.e., Aceto, M. Beverages, 2018, 4, 23).
Lines 92-93: This line can be added to the end of the previous paragraph. Please, clarify “…general and specific studies…”.
There is a lack of understanding in “Sample preparation” sub-section. Please, clearly and briefly describe the various preparation protocols of the samples according to their matrices, if this is the aim of this sub-section. It is suggested to use present tense to describe protocols since they are still used in ICP-MS analysis.
Line 95: Please, clarify the meaning of the following phrase “…with different levels…”. Line 96: HNO3 or HNO3/H2O2 are commonly used in acid digestion protocol. There are cases that HCl or HF are used depending on the sample and the metals we are interested in.
Line 96: Please, correct uses with used.
Line 99: Please, correct inclusive with including.
Line 100: I don’t understand the meaning of the use of the word “Here,” in this sentence since it is a review article.
Lines 105-107: Microwave assisted digestion method for sample preparation is not used only to dissolve heavy metals. Please, rephrase.
3. PCA TOOLS
Please, add references to this section in order to support your statements. It is suggested to better organize the structure of the last four paragraphs (Lines 148-175) of the section. There is a constant meaning repetition on when this software packages are used.
Line 114: Please, correct “…geographical origin…”.
Line 115: There is an extra space before the “While…”.
Line 118 and 119: Please, correct the grammar “…concentrates on +ing…”.
Line 119: Please, correct “…disregards…”.
Line 122: Figure 1 is not mentioned in the main manuscript.
Line 125: In the phrase “…it could reduce dimension…”, please check if you need to replace the word dimension to dimensionality or dimensions.
Line 125: Please, add the following: “…by using main principal components (PCs)…”.
Line 134: Please, add the full name before the abbreviation ML.
Line 144: Please, replace performanced to performed.
Line 146: Please, replace colon (:) to period (.).
Table 3: Please, add the web sites are mentioned to the first column of the table as references at the end of the manuscript.
Line 160: Figure 2 is not mentioned in the main manuscript.
4. APPLICATIONS OF ICP-MS COMBINED WITH PCA FOR DETERMINING THE ORIGIN OF AGRICULTURAL PRODUCTS
Lines 186-196: Please, add references to this section in order to support your statements.
Line 197: There is no Table 5.
5. THE NECESSITY OF STANDARDIZATION FOR GEOGRAPHICAL TRACEABILITY METHODS
Generally, a well-written section. It is suggested:
· Please, move figure 3 after the keys which are referred with bullets.
· It would be more consistent with the structure of a Systematic Review the proposed essential steps to be a strong outcome of the previous literature research on the topic. Consider to replace the phrase “…we propose…”, in line 235, to something more general.
Line 236: Please, correct Figure 2 with Figure 3.
6. CONCLUSIONS
The conclusion section should be modified, underlining the advantages of using ICP-MS analysis combined with the PCA statistical tool for the determination of geographical origin of plant-based products. It is necessary to clearly answer the question WHY using ICP-MS with PCA has to be a standardized approach for geographical traceability of plant-based products and which is the uniqueness of this techniques.
Author Response
Dear Respected Reviewer,
Thank you very much for your kindness comments.
The comments of reviewers were responsed point by point in the red color texts as below:
Comments of Reviewer 1:
GENERAL COMMENTS
The manuscript consists an effort to review most of research about using ICP-MS combined with the PCA for the geographical origin determination of plant-based products, and simultaneously convince their readers of the necessity this
It would be better if the authors reorganize the structure of the article clearly showing the discussion section. Please, consider sections 2 and 3 to be under one section of Methods, while sections 4 and 5 to be the Discussion, for instance.
The paper was re-structured with sections of Methods (including sub-sections 2 and 3) and Discussion (involve sub-section 4 and 5) as the reviewer’s comments
It is recommended that authors should enhance and further explain their statements by giving and discussing few examples from the literature. This is a part which is missing from the review.
Table 4 shows several examples of the applying ICP-MS and PCA in Plant Food Traceability Studies
Please, add references to the manuscript and check out the grammar and syntax of the manuscript.
Several references were added into the manuscript, and the grammar and syntax of the manuscript were double checked and revised in suitable forms.
Please, check out the reference part. Not all of them are written in the appropriate type for Foods Journal.
All references were double checked and revised in suitable format following the requirement of the FOODS journal
TITLE
Line 1: Please replace “Communication” with “Review”.
The word “communication” was replaced by “review” in the Line 1.
Line 3: I would suggest to replace the slash (/) in the abbreviation of ICP/MS with a hyphen (-) which is the most appropriate. Please correct that in all cases throughout the manuscript. It is correctly written in lines 73, 80, 82. Consider to add the full name of ICP-MS before its abbreviation.
The slash (/) in the abbreviation of ICP/MS was replaced with the hyphen (-) as ICP-MS. The full name of ICP-MS was added before its abbreviation in the title
Line 3: Consider to add the abbreviation in brackets of (PCA) after the full name.
The full name of PCA was added in the Line 3
Line 4: Correct the superscripts in the first name (*12 to 1,2,*)
The superscripts in the first name was recorrected as 1,2,*
ABSTRACT
Line 11: Since ICP-MS and PCA are referred for the first time to the text, the full names should be written and the abbreviations should be in brackets.
The full names of ICP-MS and PCA were added, and their abbreviations were put in the brackets
Line 13: Please, check out the grammar of this sentence.
The sentence was recorred as “Recent studies selected and applied ICP-MS analytical method and PCA in plant food geographical traceability”
Lines 19-21: “Furthermore, the paper proposes the main steps needed to establish the standard for this traceability method, including the development of guidelines and quality control measures.” Please, confirm if the authors are able to propose in a systematic review. The main purpose of a systematic review is to summarize all the literature research in responding to a specific research question. The answer should be extracted and supported by the studies of the literature.
This sentence was rewriten as: “Furthermore, the main steps needed to establish the standard for this traceability method were reported, including the development of guidelines and quality control measures, which play a pivotal role providing the authentic product information through each stage of production, processing and distribution for consumers and authority agencies. This might be the base to establish the standards for examination and control the quality of foods in the markets, ensuring safety for consumers”.
- INTRODUCTION
Lines 35-37 (+Table 1): Are those studies that investigate multi-elemental analysis in plant-based foods using PCA statistical method? Please, clarify.
Yes. You are right, the phrase word “using PCA statistical method” was added into this sentence
Lines 54-57: Please, check out the grammar of this sentence.
This systematic review aims to gather research on the application of ICP-MS combined with PCA as a technique to establish the authenticity of the geographical origin of plant-based foods, which could be considered to promulgate as an international standard.
Line 55: Please, correct “…geographical origin…”.
“…. the geographic origin….” was recorrected by “…the geographical origin….”
Line 63: Please, add a reference: “Currently, different types of 62 standards exist for agricultural and food products [Ref.], but standards for determining the geographical origin have not yet been widely adopted.”
The references were added in this sentence:
Eva-Marie Meemken , Christopher B. Barrett, Hope C. Michelson, Matin Qaim, Thomas Reardon and Jorge Sellare. Sustainability standards in global agrifood supply chains. Nature food. https://doi.org/10.1038/s43016-021-00360-3; Steven Jaffee and Spencer Henson. Standards and Agro-Food Exports from Developing Countries: Rebalancing the Debate. https://www.researchgate.net/publication/23722950
Lines 65-70: Please, add references.
The below reference was added into these lines.
Alfredo Cartoneaand Paolo Postiglione. Principal component analysis for geographical data:the role of spatial effects in the definition of compositeindicators. Spatial Economic Analysis, 2021, 16(2), 126–147. https://doi.org/10.1080/17421772.2020.1775876
- MULTIELEMENTAL AND ACCURACY ANALYSIS
This section is suggested to be rewritten. It is a section which needs major revisions. Please, try to be more precise and clarifying on both sub-sections (Multielement analysis, Sample preparation). Please, add references to this section in order to support your statements.
These sub-sections were revised and several references were added into the manuscript
In “Multielement analysis” sub-section, a brief explanation of why there is a need to investigate the elements for geographical discrimination, why and when are we interested in elements and their isotopes, why are there elements are not recommended and why are there elements that cannot be measured.
The sub-section “Multielement analysis” was added with more information to explain the need of investing the elements and isotopes for geographical discrimination as below:
Generally, trace elements represent the geographical tracer in a specific soil condition, which are absorbed via the roots and transferred to various parts of plant. The distribution of trace elements reflects the elemental signature of the soil origin. In addition, the isotope ratios of the elements show the linkage between products and soil characteristics. Particularly, isotopes of heavy metals have been considered as the most suitable for tracing a plant-based food ‘s origin. However, the isotopes of light elements such as hydrogen, nitrogen, oxygen, and sulfur are considered as reliable indicators of food authentication, however, the ratio of these elements are too variable to serve as tracers of the soil where a product produced (Emma et al., 2022; Drivelos et al., 2012; Rossmann et all., 2001).
References:
Emna G. Nasr, Ekaterina N. Epova, Alberto de Diego, Radhia Souissi, Mohamed Hammami, Houyem Abderrazak and Olivier F. X. Donard. Trace Elements Analysis of Tunisian and European Extra Virgin Olive Oils by ICP-MS and Chemometrics for Geographical Discrimination. Foods 2022, 11, 82. https://doi.org/10.3390/foods11010082.
Drivelos, S.A.; Georgiou, C.A. Multi-element and multi-isotope-ratio analysis to determine the geographical origin of foods in the European Union. TRAC Trends Anal. Chem. 2012, 40, 38–51.
Rossmann, A. Determination of stable isotope ratios in food analysis. Food Rev. Int. 2001, 17, 347–381.
Line 80: Please, correct the following “The elements are not recommended…”
This sentence was corrected according to the above comment
Line 82: Please, correct the following “The elements cannot be measured …”
This sentence was corrected according to the above comment
Line 85: Please, try to keep a consistent Table 2 (make all elements in Bold etc.). Please, move the table after the last paragraph.
All elements were made in bold. And the Table 2 was moved to the last paragraph
Line 86-87: The following statement is not true: “...there are elements of the Lanthanum family that are not present in food samples...”. This information is misleading. There are studies which investigate the present of Lanthanides in plant-based foods (i.e., Aceto, M. Beverages, 2018, 4, 23).
This sentence was rewritten as: Out of the remaining 58 elements, the rare earth elements (REEs) can act as geochemical markers, however, less information using REEs in foodstuff traceability
Maurizio Aceto, Federica Bonello, Davide Musso, Christos Tsolakis, Claudio Cassino, and Domenico Osella. Wine Traceability with Rare Earth Elements. Beverages 2018, 4(1), 23; https://doi.org/10.3390/beverages4010023.
Lines 92-93: This line can be added to the end of the previous paragraph. Please, clarify “…general and specific studies…”.
The line 92-93 was added to the end of the previous paragraph in more clear meaning: “These 36 elements are commonly analyzed using ICP-MS methods for the elemental or multi-elemental determination in food traceability”
There is a lack of understanding in “Sample preparation” sub-section. Please, clearly and briefly describe the various preparation protocols of the samples according to their matrices, if this is the aim of this sub-section. It is suggested to use present tense to describe protocols since they are still used in ICP-MS analysis.
The “sample preparation” sub-section was rewritten as below: Solid samples are digested in strong and hot acid conditions, such as: HNO3, HNO3/HCl, HNO3/H2O2, or HNO3/HF…, which depend on the specific matrices and the metals are studied. In general, samples are commonly digested with pure HNO3 (65-70%) on microwave oven, then diluted with ultra-pure water (da Silva, I.J.S.; Lavorante, A.F.; Paim, A.P.S.; da Silva, M.J. Microwave-assisted digestion employing diluted nitric acid for mineral determination in rice by ICP OES. Food Chem. 2020, 319, 126435). There were various methods converting solid samples into aerosols, inclusing: electrothermal vaporization (ETV), laser ablation (LA), microwave-assisted digestion (MAD), spark ablation… The samples were then transported to the plasma by an inert gas. In these techniques, the ETV analysis method is used for combustible samples while the spark ablation is applied for conducting samples in sampling large spots with a diameter of 1-3 mm. Conversely, LA microanalysis technique using high-irradiance (UV) lasers to measure very small spots (2–750 µm in diameter) on almost solid samples as well as useful for tracking element distribution in a sample. MAD method for sample preparation is suitable analysis by ICP-MS, inductively coupled plasma atomic emission spectrometry (ICP-AES), graphite furnace atomic absorption spectrophotometry (GFAA), flame atomic absorption spectrophotometry (FLAA).
Line 95: Please, clarify the meaning of the following phrase “…with different levels…”. Line 96: HNO3 or HNO3/H2O2 are commonly used in acid digestion protocol. There are cases that HCl or HF are used depending on the sample and the metals we are interested in.
The phrase “… with different levels…” in the sentence was removed out of “sample preparation” sub-section
Line 96: Please, correct uses with used.
The word “uses” in the sentence was removed out of “sample preparation” sub-section
Line 99: Please, correct inclusive with including.
The word “inclusive” in the sentence was removed out of “sample preparation” sub-section
Line 100: I don’t understand the meaning of the use of the word “Here,” in this sentence since it is a review article.
The word “Here” was replaced by the phrase “In these techniques”
Lines 105-107: Microwave assisted digestion method for sample preparation is not used only to dissolve heavy metals. Please, rephrase.
These lines were rephrased as: MAD method for sample preparation is suitable analysis by ICP-MS, inductively coupled plasma atomic emission spectrometry (ICP-AES), graphite furnace atomic absorption spectrophotometry (GFAA), flame atomic absorption spectrophotometry (FLAA).
- PCA TOOLS
Please, add references to this section in order to support your statements. It is suggested to better organize the structure of the last four paragraphs (Lines 148-175) of the section. There is a constant meaning repetition on when this software packages are used.
Line 114: Please, correct “…geographical origin…”.
It was corrected
Line 115: There is an extra space before the “While…”.
The space was deleted
Line 118 and 119: Please, correct the grammar “…concentrates on +ing…”.
It was corrected following the comment, as: “LDA concentrates on finding a feature subspace which helps to enhance the separability between groups…”
Line 119: Please, correct “…disregards…”.
It was corrected in the manuscript
Line 122: Figure 1 is not mentioned in the main manuscript.
Figure 1 was mentioned via the sentence: “Figure 1 shows the different number of research publications using PCA, LDA, KNN and HCA multivariate statistical methods for geographical origin determination”.
Line 125: In the phrase “…it could reduce dimension…”, please check if you need to replace the word dimension to dimensionality or dimensions.
The word “dimension” was replaced by “dimensionality”
Line 125: Please, add the following: “…by using main principal components (PCs)…”.
The word “principal” was added into this sentence
Line 134: Please, add the full name before the abbreviation ML.
The full name was added before the abbreviation ML
Line 144: Please, replace performanced to performed.
The word “performanced” was replaced by performed
Line 146: Please, replace colon (:) to period (.).
The colon (:) was replaced by the period (.)
Table 3: Please, add the web sites are mentioned to the first column of the table as references at the end of the manuscript.
The websites mentioned to the first column of the table were listed as references at the end of the manuscript: “Note: the websites of softwares were listed corresponding as: XLSTAT (xlstat.com), R (r-project.org), Minitab (minitab.com), Matlab (mathworks.com), SPSS (ibm.com), Weka (cs.waikato.ac.nz), SIMCA (sartorius.com)”.
Line 160: Figure 2 is not mentioned in the main manuscript.
Figure 2 was mentioned in the end of the last paragraph of sub-section “PCA tools”
- APPLICATIONS OF ICP-MS COMBINED WITH PCA FOR DETERMINING THE ORIGIN OF AGRICULTURAL PRODUCTS
Lines 186-196: Please, add references to this section in order to support your statements.
Several references were added into this section as below:
Jessica L. Johnson, Donna Adkins, and Sheila Chauvin. A Review of the Quality Indicators of Rigor in Qualitative Research. Am J Pharm Educ. 2020 Jan; 84(1): 7120. doi: 10.5688/ajpe7120.
Samantha Islam, Jonathan M Cullen. Food Traceability: A Generic Theoretical Framework. Food Control 123(3):107848].
Line 197: There is no Table 5.
It is recorrected with Table 4
- THE NECESSITY OF STANDARDIZATION FOR GEOGRAPHICAL TRACEABILITY METHODS
Generally, a well-written section. It is suggested:
- Please, move figure 3 after the keys which are referred with bullets.
Figure 3 was moved after the keys which are referred with bullets
- It would be more consistent with the structure of a Systematic Review the proposed essential steps to be a strong outcome of the previous literature research on the topic. Consider to replace the phrase “…we propose…”,in line 235, to something more general.
The phrase “…we propose…” was replaced by “there were several….”
Line 236: Please, correct Figure 2 with Figure 3.
Figure 2 was corrected by Figure 3
- CONCLUSIONS
The conclusion section should be modified, underlining the advantages of using ICP-MS analysis combined with the PCA statistical tool for the determination of geographical origin of plant-based products. It is necessary to clearly answer the question WHY using ICP-MS with PCA has to be a standardized approach for geographical traceability of plant-based products and which is the uniqueness of this techniques.
The conclusion section was rewritten as below:
The present review summarizes the research on the application of ICP-MS and PCA in the geographical origin authentication of agricultural products. Consequently, ICP-MS is a robust, accurate, and highly sensitive technique for determining the inorganic elements in food substance, whereas PCA can reduce dimensions, speed up machine learning processes, prevent data overflow and reduce noise. The combination of ICP-MS and PCA can be considered as a powerful tool and a standardized approach in authenticating and certificating the geographical origin of plant-based foods, which plays an important role in protecting quality products. In addition, this might be the base for producers making decisions to enhance the effectiveness of the certification of their products to match the demand of consumers in the markets.

Reviewer 2 Report
The manuscript “Towards a standardized approach for geographical traceability 2 of plant foods using ICP/MS and principal component analysis” needs a little bit of improvement which should be addressed. In my opinion, the first section, Introduction, should be improved and the reference in the manuscript should be correctly numbered.
The entire sample preparation section should be written using the correct verb tense since the paper presents a review and no measurements or preparation were performed.
Figure 2 caption should include the source and I would recommend to the authors to improve table 4 by adding at least 2 references per plant.
The aim of study should be revised and presented in separate paragraph: Authors are encouraged to prepare the hypothesis and try to describe novelty of this research.
Author Response
Dear Respected Reviewer,
Thank you very much for your kindness comments.
The comments of reviewers were responsed point by point in the red color texts as below:
Comments of Reviewer 2:
The manuscript “Towards a standardized approach for geographical traceability 2 of plant foods using ICP/MS and principal component analysis” needs a little bit of improvement which should be addressed. In my opinion, the first section, Introduction, should be improved and the reference in the manuscript should be correctly numbered.
The manuscript was improved, and the references were re-numbered in the order
The entire sample preparation section should be written using the correct verb tense since the paper presents a review and no measurements or preparation were performed.
The grammars in the manuscript were double checked and revised in suitable tense
Figure 2 caption should include the source and I would recommend to the authors to improve table 4 by adding at least 2 references per plant.
Several references were added into the Table 4 as comments, however, we could not found any more publications of a few plant foods such as: jackfruits, maize and parkchoi in the Table 4.
The aim of study should be revised and presented in separate paragraph: Authors are encouraged to prepare the hypothesis and try to describe novelty of this research.
The aim of study was rivised and present in the abstract of the manuscript

Round 2
Reviewer 1 Report
The manuscript has been considerably improved after the revisions.
Line 12: Please, delete “ICP-MS” in the sentence “…ICP-MSInductively coupled plasma mass spectrometry (ICP-MS)”
Line 96: “The elements that cannot …measurable…” Verb is missing
Lines 119-122: Please, check the fond of the sentence.
Line 136: It would be better if figure 1 was placed after the paragraph in which figure 1 is mentioned. After lines 139-148.
Lines 176-177: It would be better if figure 2 was placed after the last paragraph of the section.
Line 257-258: There is a two-line space between the paragraphs.
Lines 270-271: Please, delete the bullets.
In figure 3, please correct “ICP/MS” to “ICP-MS”.
Author Response
Dear Respected Reviewer,
Thank you very much for your kind comments.
We already responded for the comments of reviewers point by point with the red text as below:
Comments and Suggestions for Authors
The manuscript has been considerably improved after the revisions.
Line 12: Please, delete “ICP-MS” in the sentence “…ICP-MS Inductively coupled plasma mass spectrometry (ICP-MS)”
It was deleted
Line 96: “The elements that cannot …measurable…” Verb is missing.
This sentence was rewritten as “The elements that cannot measured by ICP-MS….”
Lines 119-122: Please, check the fond of the sentence.
The LA microanalysis technique uses high-irradiance (UV) lasers to measure very small spots (2–750 µm in diameter) on almost solid samples whilst MAD method was applied for sample preparation process in the analysis by ICP-MS, inductively coupled plasma atomic emission spectrometry (ICP-AES), graphite furnace atomic absorption spectrophotometry (GFAA), flame atomic absorption spectrophotometry (FLAA).
Line 136: It would be better if figure 1 was placed after the paragraph in which figure 1 is mentioned. After lines 139-148.
Figure 1 was placed after the paragraph in which figure 1 is mentioned.
Lines 176-177: It would be better if figure 2 was placed after the last paragraph of the section.
Figure 2 was placed after the last paragraph of the section.
Line 257-258: There is a two-line space between the paragraphs.
It was double checked without two-line space between the paragraphs.
Lines 270-271: Please, delete the bullets.
The bullets in lines 270-271 were deleted.
In figure 3, please correct “ICP/MS” to “ICP-MS”.
The “ICP/MS” in figure 3 was correct by “ICP-MS”.
Reviewer 2 Report
The authors did not answer or address to some of the previously submitted comments, so from my point of view no improvements were made.
Figure 2 caption should include the source and I would recommend to the authors to improve table 4 by adding at least 2 references per plant. Authors are encouraged to describe novelty of this research.
Author Response
Dear Respected Reviewer,
Thank you very much for your kind comments.
We already responded for the comments of reviewers point by point with the red text as below:
Comments and Suggestions for Authors
The authors did not answer or address some of the previously submitted comments, so from my point of view no improvements were made.
Figure 2 caption should include the source and I would recommend to the authors to improve table 4 by adding at least 2 references per plant. Authors are encouraged to describe the novelty of this research.
Thank you very much for your kind comments.
- The data in Figure 2 was summarized from various researches, which were published in ScienceDirect system, so it did not have specific source.
- Actually, under your comments in the first round of reviewing, we already added one more reference per plant in the Table 4 in the previous revised manuscript (including: cherry, almonds, sesame, wheat, potato, asparagus, cabbages, and mushroom) in comparison with that of the smutted manuscript. Those references were provided as below, involved:
Spyridon, P.; Artemis, L.; Ioanna, K.; Stavros, K.; Anastasia, B.; Chara, P.; Michael, G.K. Physicochemical, Spectroscopic and Chromatographic Analyses in Combination with Chemometrics for the Discrimination of Four Sweet Cherry Cultivars Grown in Northern Greece. Foods. 2019, 8(10), 442.
Diana, A.; Santino, O.; Andrea, P.; Salvatore, B. Discrimination of almonds (Prunus dulcis) geographical origin by minerals and fatty acids profiling. Nat. Prod. Res. 2015 http://dx.doi.org/10.1080/14786419.2015.1107559.
Young, H.C.; Chae, K.H.; Misun, K.S. O. J.; Juseong, P.; Young, H.O.; Joong-Ho, K. Multivariate analysis to discriminate the origin of sesame seeds by multi-element analysis inductively coupled plasma-mass spectrometry. Food. Sci. Biotechnol. 2017,26(2) 375-379.
Hongyan, L.; Yimin, W.; Yingquan, Z.; Shuai, W.; Senshen, Z.; Boli, G. The effectiveness of multi-element fingerprints for identifying the geographical origin of wheat. Int. J. Food. Sci. Technol. 2017. doi:10.1111/ijfs.13366.
Franco, D.G.; Antonella, D.S.; Mario, G. Determining the Geographic Origin of Potatoes Using Mineral and Trace Element Content. J. Agric. Food Chem. 2007, 55, 860−866.
Yong, K.K.; Yong, K.K.; Yeon-Sik, B.; Kwang-Sik, L.; wang-Sik, L.; Geum –Sook, H. An integrated analysis for determining the geographical origin of medicinal herbs using ICP-AES/ICP-MS and 1H NMR analysis. Food. Chem. 2014, 161, 168–175. DOI: 10.1016/j.foodchem.2014.03.124.
Yeon-Sik, B.; Woo-Jin, S.; Mukesh, K.G.; Youn-Joong, J.; A-Reum, L.; Chang-Soon, J.; Yong-Pyo, L.; Gong-Soo, C.; Kwang-Sik, L. Determining the geographical origin of Chinese cabbages using multielement composition and strontium isotope ratio analyses. Food. Chem. 2012, 135(4), 2666-2674.
Segelke, T.; von Wuthenau, K.; Neitzke, G.; Müller, M.-S.; Fischer, M. Food Authentication: Species and Origin Determination of Truffles (Tuber spp.) by Inductively Coupled Plasma Mass Spectrometry and Chemometrics. J. Agric. Food Chem. 2020, 68, 14374–14385.
- In addition, there was a reference adding for jackruit as: “Nadia, D. Identification of the geographical origin of jackfruit (Artocarpus heterophyllus Lam.) through multielemental fingerprinting using ICP. Thesis of School of Life Sciences – Sikkim University, Gangtok, Sikkim, India, 2018.”
- On the other hand, we tried to find more references to support for Table 4, however, we couldn’t find more references for some types of plants such as: maize, ricinus and pakchoi.
- The novelty of this research would like to illustrate that the combination of ICP-MS and PCA can be considered a powerful tool and a standardized approach to authenticating and certifying the geographical origin of plant-based foods. The use of a standard method will reduce the time and cost of analysis and improve the efficiency of trade and circulation of goods. Furthermore, the main steps needed to establish the standard for this traceability method were reported, including the development of guidelines and quality control measures, which play a pivotal role in providing authentic product information through each stage of production, processing, and distribution for consumers and authority agencies. This might be the basis for establishing the standards for examination and controlling the quality of foods in the markets, ensuring safety for consumers. This information was shown in the abstract and conclusion of this paper.
